# Knowledge and Practices in Neonatal Pain Management of Nurses Employed in Hospitals with Different Levels of Referral—Multicenter Study

**DOI:** 10.3390/healthcare9010048

**Published:** 2021-01-05

**Authors:** Hanna Popowicz, Wioletta Mędrzycka-Dąbrowska, Katarzyna Kwiecień-Jaguś, Agnieszka Kamedulska

**Affiliations:** 1Department of Obstetric and Gynecological Nursing, Medical University of Gdańsk, 80-211 Gdańsk, Poland; hanna.popowicz@gumed.edu.pl; 2Department of Anaesthesiological Nursing and Intensive Care, Medical University of Gdańsk, 80-211 Gdańsk, Poland; katarzyna.kwiecien-jagus@gumed.edu.pl; 3Department of Biopharmacy and Pharmacokinetics, Medical University of Gdańsk, 80-416 Gdańsk, Poland; agnieszka.kamedulska@gumed.edu.pl

**Keywords:** neonatal pain, knowledge, nurses, midwives, prophylaxis, pain management, neonate, neonatal intensive care unit

## Abstract

Background: One of the key elements of patient care is the relief and prevention of pain sensations. The importance of pain prevention and treatment has been emphasized by many international organizations. Despite the recommendations and guidelines based on evidence, contemporary research shows that the problem of pain among patients in neonatal intensive care units (NICUs) in various centers is still an important and neglected problem. Aim: The aim of this study was to assess the level of knowledge of the medical personnel and their perception of the issue of pain in neonatal patients. Methods: A quantitative descriptive study carried out in 2019. The study used a nurses’ perceptions of neonatal pain questionnaire. Results: A total of 43 Polish hospitals and 558 respondents participated in the project. 60.9% (*n* = 340) and 39.1% (*n* = 218) of respondents were employed in secondary and tertiary referral departments, respectively. Conclusion: Our analyses indicate that despite the availability of pain assessment tools for neonatal patients, only a few centers use standardized tools. The introduction of strategies to promote and extend the personnel’s awareness of neonatal pain monitoring scales is necessary.

## 1. Introduction

In the 1990s, the three-stage perinatal care system was introduced in Poland [1]. As envisaged, care and adequate therapy should depend on the condition of the neonate. The first stage is responsible for providing care for healthy neonates, and short-term specialist care, if needed. Term-born and preterm-born children who require more specialist care are admitted to intensive care units. Regardless of the referral level, a patient admitted to such a department requires continuous supervision, therapy for circulatory and respiratory disorders, the introduction and continuation of life-supporting procedures, the monitoring of vital parameters, and adequate diagnostic and professional intensified nursing care [2,3]. Pain relief and prevention are key elements in patient care. The importance of pain prophylaxis and therapy was emphasized by multiple international organizations, such as the World Health Organization, the Academy of Pediatrics (APP) and the Canadian Pediatric Society (CPS), and the Polish Neonatal Society (PTN) [4,5,6]. Although many evidence-based recommendations and guidelines have been published, recent studies demonstrate that the issue of pain in patients admitted to neonatal intensive care units is still an important and neglected problem in many hospitals. Moreover, several authors suggest that pain monitoring and therapy, regardless of the developmental age of the child, remains insufficient [7,8,9]. Scientific evidence indicates that the subcortical and cortical centers responsible for pain experiences are already sufficiently developed in the foetus [10]. Long-term, untreated pain during hospitalization has an impact on concentration, cognitive functions and the occurrence of emotional and physical disturbances [10]. This applies to extremely preterm-born infants in particular, whose stay at the neonatal intensive care unit not only involves a spectrum of painful diagnostic and therapeutic procedures of varying degrees, but is also significantly longer due to their condition [10,11,12]. Neonates experience an average of 12–16 medical procedures per day, most of which cause mild or moderate pain. Apart from these, it is also worth mentioning heel prick, venous catheter insertion, venous blood collection, respiratory tract suctioning or intubation [13,14].

The prophylaxis of pain and discomfort associated with the necessary medical procedures includes pharmacological (paracetamol, morphine, fentanyl) and non-pharmacological interventions. The latter involves non-nutritive sucking, breast milk, the application of glucose on the tongue, kangaroo mother care and wrapping. Such methods have turned out to be efficient and well-tolerated by children in the prevention of mild pain, and for moderate and acute pain relief [4,15]. The medical personnel of neonatal intensive care units (NICUs), including nurses and midwives, are responsible for the use of both non-pharmacological and pharmacological methods. The identification of nociceptive impulses plays a key role in undertaking successful preventive measures, as neonates are not capable of verbalizing their pain. It is recommended that pain intensity assessment requires the use of validated scales based on behavioral and/or physiological parameters. Such a tool should be selected with regard to the age of the child [4,15,16]. The results of recent studies seem to confirm that the medical personnel of the NICU have gained adequate knowledge about neonatal pain and the need for its systematic monitoring [8,17,18,19]. On the other hand, some authors indicate that despite binding guidelines or detailed recommendations, this is only general (basic) knowledge. Other researchers demonstrate multiple inadequate implementations of the guidelines in terms of pain therapy. These include the lack of a continuous and regular pain intensity assessment, the incomplete introduction of non-pharmacological methods of pain relief and the low efficiency of pain therapy. This may cause psychological, behavioral, metabolic, mental and hormonal reactions [8,17,18,19].

### Aim

The objective of this study was to evaluate the knowledge and understanding of medical staff towards the pain of neonates.

## 2. Methods

### 2.1. Design

The research was conducted with the use of a descriptive method. The project was carried out in Polish neonatal intensive care units.

### 2.2. Participants

The project was carried out in 2019. The study group included 558 medical professionals (midwives, nurses) employed in neonatal intensive care units classified as secondary and tertiary referral departments (a total of 43 Polish hospitals).

### 2.3. Data Collection

The study used a nurses’ perceptions of neonatal pain questionnaire, which included 36 Likert scale and 2 open questions [20].

The questionnaire required the selection of one of the answers in a 5-point Likert scale, where 1 represented “strongly disagree” and 5 referred to “strongly agree”. The author of the original version of the questionnaire gave her consent to use this tool in our project. The questionnaire was constructed to focus on 5 aspects of perceptions of neonatal pain by nurses: (1) knowledge and beliefs; (2) use of assessment tools; (3) use of pharmacological and non-pharmacological interventions; (4) guidelines/protocols and family involvement; and (5) barriers and strategies. This research tool was translated by two independent translators. The questionnaire was supplemented with socio-demographic questions, including the place of residence and general work experience. The questions concerning race and ethnicity were removed from the Polish version of the questionnaire, as the above-mentioned information does not correspond to Polish conditions—Poland is among the states with the lowest percentages of non-national populations in the European Union, with only 0.4% of the Polish population represented by immigrant populations [21].

Similarly, the area of professional practice was limited to two answers, i.e., the secondary and tertiary referral levels of medical care. The number of distractors concerning age and professional experience in the NICU were reduced, and the answers regarding education level were adjusted to the national education system. The open question concerning barriers in pain management was changed into a closed question, so that the respondent could choose the three most important barriers in her/his opinion. A question was added in the part of the questionnaire concerning the tools used for pain monitoring, requiring the respondent to name the tool/scale of pain assessment which is used in the department participating in the project. The statistical analysis of internal consistency with Cronbach’s alpha was 0.888. The internal reliability of the tool indicates that the Polish version of the scale is reliable.

### 2.4. Study Procedure

One hundred and twenty hospitals with neonatal intensive care units (secondary and tertiary referral levels) were invited to participate in this project. Out of 52 hospitals, only 43 were included in the study procedure although all of the medical centers agreed to participate in the project. The questionnaire package was delivered to all the medical facilities which gave their official consent. In some of the facilities, medical staff refused to participate in the research, which is why those hospitals were not part of the further analysis.

### 2.5. Inclusion and Exclusion Criteria

The primary inclusion criteria in this project were: workplace—NICU, secondary and tertiary referral level, profession—professionally active and registered nurses and midwives, actively working in the above-mentioned departments, anonymous informed consent of the respondent to participate in the project, and reception of official consent from the directors/chairmen of the participating hospitals. Subjects who met any of the following exclusion criteria did not participate in the study: profession other than nurse/midwife, not registered nurse or midwife, nurse or midwife who is not employed in the NICU, lack of consent from the hospital management, lack of consent to participate in the study.

### 2.6. Statistical Analysis

All statistical calculations were carried out using the IBM SPSS 23 (IBM, Armonk, NY, USA) statistical package and an Excel 2016 spreadsheet. Qualitative variables were presented as numbers and percentages, while quantitative variables were characterized using the arithmetic mean and standard deviations. The significance of differences between more than two groups was tested using the Kruskal–Wallis test and the single factor analysis of variance (ANOVA). The significance of differences between two groups was tested using the Student’s *t*-test. In all calculations, an adjusted *p*-value (Bonferroni correction) less than 0.05 was considered as the level of significance.

## 3. Results

### 3.1. Sociodemographic Characteristics

A total of 43 Polish hospitals participated in the study and we received 558 correctly completed questionnaires (60.9% and 39.1% of the respondents were employed in secondary and tertiary referral departments, respectively). Cities below 50 thousand inhabitants were the place of residence for most of the group of the studied subjects, while the fewest individuals from the group lived in cities of 50,000 to 100,000 citizens. The highest number of respondents were 31 to 50 years old, with professional experience of more than 10 years of working in the NICU (64.9%) and had secondary education (graduated a medical school of nursing or medical college—37.5%). The detailed socio-demographic characteristics of the study group are presented in Table 1.

### 3.2. Nurses’ Perceptions of Neonatal Pain

A detailed summary of 5 aspects of the nurses’ perceptions of neonatal pain is presented in Table 2.

### 3.3. Nurses’ Knowledge and Beliefs about Neonatal Pain

Although a significant number of participants in the study group (M 4.66; SD ± 0.61) realized that neonates will remember painful experiences, the majority of those (M 3.71; SD ± 1.03) also believed that neonates, and especially preterm infants are less sensitive to pain than older children and adults because of the immature immune system. Moreover, the respondents agreed with the statement that pain experienced in the neonatal period has long-term adverse effects (M 4.04; SD ± 0.90). However, they considered that preterm children were less susceptible to pain than older children (M 3.9; SD ± 1.14), but that the former will remember painful experiences (M 3.26; SD ± 1.02).

### 3.4. Use of Pain Assessment Tools

The statistical analysis demonstrated that the arithmetic mean for the use of neonatal pain assessment tools ranged from M 2.24 to M 3.49. The highest rate was observed in subjects who felt confident in the identification of pain indicators using the tools which are available in their unit (M 3.49; SD ± 0.93), while the lowest rate was presented by the respondents (M 2.24; SD ± 2.49) who attended an introductory course in pain identification/evaluation when newly employed in the intensive care unit. The arithmetic mean regarding regular education in neonatal pain management was M 2.91 (SD ± 1.14), and an average of 2.67 (SD ± 1.06) respondents regularly use neonatal pain assessment tools [15,16].

Furthermore, the respondents stated that the assessment tool used in their department provides an accurate evaluation of neonatal pain (M 2.76; SD ± 1.03). The results of this study demonstrate that the most frequently used neonatal pain assessment tools among nurses and midwives ware (in decreasing order of frequency): Cry, Requires increased oxygen administration, Increased vital signs, Expression, Sleeplessness (CRIES)—13.3% (*n* = 74), Neonatal-Pain, Agitation and Sedation Scale (N-PASS)—13.1% (*n* = 73), Neonatal Infant Pain Scale (NIPS)—8.8% (*n* = 49). More than 55.2% of the studied subjects (*n* = 308) did not use the recommended tools for pain monitoring in this group of patients.

### 3.5. Use of Pharmacological and Non-Pharmacological Interventions

A significant number of the participants in the study group held the opinion that it is the nurses/midwives’ duty to promote issues associated with pain prophylaxis and management in their department (M 4.03; SD ± 0.73). Some respondents (M 2.82; SD ± 1.09) did not feel adequately trained by their employer in both pain prophylaxis and the forms of its management in paediatric patients.

The respondents agreed that pharmacological and non-pharmacological interventions are necessary and efficient in pain relief, despite the fact that many invasive procedures may be carried out quickly. The analysis of their responses demonstrates that neonatal pain is indeed treated in most of the centers that participated in this study (M 3.58; SD ± 0.95).

### 3.6. Guidelines/Protocols for Pain Management

More than 40.7% of the respondents (*n* = 227) stated that they know the guidelines/protocols concerning pain management in their departments (M 3.17; SD ± 0.96). Moreover, the great majority of nurses/midwives consider them understandable and clear (M 3.04; SD ± 0.99). The studied medical personnel introduce and use non-pharmacological pain relief methods in their everyday practice, in the event that pharmacological methods are not efficient enough (M 3.65; SD ± 0.86). It is not a problem for the respondents to discuss neonatal pain therapy with physicians (M 3.10; SD ± 1.05).

### 3.7. Family Involvement and Cultural Aspects of Pain Management

The nurses/midwives participating in the study were aware that pain experienced by neonates born at term and preterm children has an impact on their parents’ emotional status (M 4.36; SD ± 0.57). The studied personnel were aware that legal guardians expect pain relief actions to be undertaken (M 4.08; SD ± 0.72), and that they should be present during painful procedures (M 3.63; SD ± 1.02).

More than 90% of the participating subjects agreed that kangaroo mother care is an efficient non-pharmacological method for neonatal pain relief (M 4.14; SD ± 0.69).

### 3.8. Strategies and Barriers

Five levels of awareness of pain prophylaxis, management and monitoring were distinguished in personnel: average—56.3% (*n* = 314), high – 19.7 % (*n* = 110), low—13.4% (*n* = 75), very low—7.2% (*n* = 40), and very high 3.4% (*n* = 19). According to the respondents, the most important barriers in effective pain management in the facility where they work include: lack or insufficient amount of training 63.1% (*n* = 352); haste, insufficient number of personnel 59.1% (*n* = 330), unwillingness to change current practices 40.3% (*n* = 225); poor practices 39.1% (*n* = 218); inadequate communication within the team, contradicting opinions concerning pain assessment and therapeutic options 32.6% (*n* = 182); no belief in the reliability of pain assessment scales 24.7% (*n* = 138); other mentioned issues: no procedures, physician’s attitude 2.3% (*n* = 13).

The respondents stated that appropriate care, which includes pain prophylaxis, requires the adequate education of medical personnel and the development of internal strategies within the facility where they are employed. Another important aspect is to change the awareness of the medical personnel both in terms of prophylaxis and methods of pain therapy, especially non-pharmacological methods.

### 3.9. Variables Associated with Knowledge and Opinions about Pain Relief and Monitoring by Nurses/Midwives

The highest usage of the pain evaluating tools (H_(2)_ = 15.79; *p* < 0.001), methods of pain relief (H_(2)_ = 30.71; *p* < 0.001) and guidelines/protocols (H_(2)_ = 29.69; *p* < 0.001) was demonstrated by subjects older than 50 years. It was significantly higher than in the study subjects aged less than 30 years and those between 31 and 50 years. No statistically significant differences were observed between other variables (*p* > 0.05).

Our analysis demonstrates that subjects with secondary education mastered the issue of neonatal pain to a very small degree (F (2.555) = 10.45, *p* < 0.001). Further analyses did not show any association between education level and: the level of knowledge of methods used in pain assessment (F = 1.50; df 2.555; *p* = 0.223), the level of knowledge of pain relief methods (F = 0.85; df 2.555; *p* = 0.428) and the level of knowledge of protocols and guidelines (F = 1.69; df 2.555; *p* = 0.185). However, the research confirmed a correlation between the education level of the studied respondents and the consideration of family aspects and parents’ opinions on the need to reduce pain in neonatal patients (F = 7.46; df 2.555; *p* = 0.001).

The interpretation of our results demonstrated statistically significantly higher knowledge of guidelines/protocols concerning neonatal pain therapy and monitoring (H_(2)_ = 10.52; *p* < 0.05) in subjects with professional experience of more than 10 years when compared with respondents with shorter professional experience. No statistically significant differences were observed between other variables and groups (*p* > 0.05).

The analysis of particular variables demonstrated that a higher level of knowledge regarding neonatal pain therapy (t_(555)_ = 3.96; *p* < 0.001) and family aspects of pain reduction t_(555)_ = 2.08; *p* < 0.05 was displayed by respondents employed in tertiary referral departments, than by those employed in secondary referral centers. Moreover, a higher level of knowledge of pain relief (t_(555)_ = 1.96; *p* = 0.050) and knowledge of protocols/guidelines (t_(555)_ = 2.70; *p* < 0.050) was also presented by subjects employed in secondary referral departments, than by those employed in tertiary referral centers. No statistically significant differences were obtained between other variables and groups (*p* > 0.05) (Table 3).

## 4. Discussion

Our study analyzed the level of knowledge of medical personnel and observations regarding pain prophylaxis and therapy in Polish neonatal intensive care units. The results clearly show that the majority of respondents have a sufficient level of knowledge of pain. However, in most of the hospitals there are no proper solutions related to pain assessment tools. This is definitely an area for future improvement. We demonstrated that although medical personnel had obtained knowledge concerning pain in this patient group, a small percentage of nurses/midwives still do not agree with the statement that long-term pain may result in a range of adverse effects. More than 50% of employed personnel were unaware that neonates undergoing necessary medical procedures would remember painful experiences. Our results were confirmed by other international researchers [20,22,23,24] who used the same original version of the questionnaire. In Poland, the project is the first one of this kind, not only because it is focused on the problem of pain in the neonatal unit, but also because the study was dedicated to the medical staff who are involved in pain management. Cong et al. and Polkki et al. [20,22,25] confirmed that the level of knowledge of medical personnel significantly correlated with age, education level and the degree of referral of hospitals in which the study was carried out. Our results demonstrated that respondents older than 50 and those employed in tertiary referral centers presented a good level of neonatal pain knowledge. However, medical staff in secondary referral care had a higher level of knowledge in the area of guidelines and protocol when compared with staff from tertiary hospitals. We conclude that unawareness of this significant issue may have an impact on the attitude of medical personnel towards actions aimed at pain prevention, and therapy and pain assessment methods in such sensitive patients as preterm neonates and severely ill children born at term. It should be emphasized that long-term chronic pain or long-lasting repetitive nociceptive impulses in this population may result in cognitive, motor, behavioral and personality disturbances [11,26,27].

The guidelines issued by scientific societies, and the Polish legal regulations based on them, define that effective pain therapy should be based on its adequate identification and on an evaluation of its intensity [5,28]. Therefore, it is indispensable to carry out adequate training on physiological and behavioral pain indicators in children. The results of our study show that the training for newly employed nurses/midwives was insufficient and that no adequate postgraduate educational programs were available.

The level of knowledge and the skills of respondents who use pain assessment scales in their routine practice are also unsatisfactory. Almost half of the studied subjects were not confident in the identification of pain indicators when required to use such scales. Similar results were obtained by Ozawa et al. [29] and Collados-Gómez et al. [30]. These authors confirmed that only 40% of nurses declared the use of tools which enable the assessment of pain intensity and the evaluation of pain perception in NICU patients [29,30]. Cong et al. [22] presented completely different results. The author and his research team stated that more than 80% of nurses used pain assessment tools. All respondents who used pain assessment scales in their routine practice were confident in their use and interpretation with neonatal patients [16].

In our project we observed that subjects aged 50 years and above used pain assessment scales more frequently than nurses/midwives of a younger age.

The COMFORT and N-PASS scales are the most commonly recommended scales for pain monitoring and the estimation of the need for analgesics for invasively ventilated neonates. However, according to the currently binding guidelines, patients experiencing acute or postoperative pain require other tools, such as the PIPP (Premature Infant Pain Profile) and CRIES scales [4,15]. The results of our study clearly indicate that pain assessment tools are very rarely used in everyday clinical practice, or they are used incorrectly. The selection of an appropriate assessment tool depends on the clinical condition of the patient requiring intensive care. No single universal scale analyzing acute/procedure-related or chronic pain has been developed for this category of patients [16,31]. The study carried out by Avila-Alvarez et al. [32] seems to confirm this statement, as these researchers showed that six completely different scales were used for pain monitoring in 10 neonatal intensive care units.

The analysis of responses of nurses/midwives working in Polish NICUs confirmed that the most commonly used scales include CRIES, N-PASS and NIPS. These results do not differ much from the observations of other authors [30,32]. Regardless of the type of measurement tool used, the percentage of personnel using it in routine clinical practice remains insufficient. This may result from inadequate training and a small number of staff responsible for carrying out such training [5]. Only 26% of respondents confirmed that the healthcare center where they are employed provides them with continuous education regarding neonatal pain therapy. Similar results were obtained by Razeq et al. [23] and Mehrnoush et al. [24]. On the other hand, Cong et al. [22] reported that more than 50% of their respondents confirmed receiving education on pain perception in such a sensitive population as neonates and being regularly trained in that field.

Another important aspect of providing care to preterm children and neonates born at term in intensive care units is the reduction or complete relief of pain. Both pharmacological and non-pharmacological interventions are available [5,33,34,35]. In our project the subjects ware confident in using pharmacological and non-pharmacological methods of pain relief. Similar results were obtained in the studies carried out by Cong et al. [20] and Mehrnoush et al. [24], where the great majority of subjects considered the combined use of both modalities necessary during invasive procedures, even if the time required to perform them was short. Mehrnoush et al. [24], however, observed that their subjects were more confident in the use of non-pharmacological methods than pharmacological ones. The non-parametric tests carried out in our study confirmed the relationship between pain, the use of relief actions, the age of medical personnel and the referral level of the hospital. A higher level of knowledge was observed in respondents over 50 years of age and those working in secondary referral centers. Furthermore, nursing personnel confirmed the efficacy of kangaroo mother care and the oral administration of sweet solutions in neonatal pain relief. These results are confirmed by the results of other scientific papers [20]. On the other hand, some authors confirm the efficacy of other non-pharmacological methods, including the administration of saccharose, wrapping and non-nutritive sucking [18,30,36].

Eighty-six percent of nurses/midwives participating in our study stated that it is the duty of the medical personnel employed in the intensive care unit to talk to and educate others on the topic of neonatal pain prevention. This may result from the high level of knowledge and awareness of medical personnel regarding pain perception. The interpretation of the obtained data confirms that approx. 40% of departments participating in our study developed evidence-based guidelines regarding the prophylaxis, monitoring and therapy of pain. Unfortunately, only 37.1% of respondents consider them clear and understandable. The presented results are slightly better than the ones reported by Mehrnoush et al. [24], who found that 29.2% of respondents claimed they had been provided with pain management protocols and 35.2% of respondents found these protocols understandable. Different results were obtained by Cong et al. [20] and Razeq et al. [23]. The detailed analysis of data demonstrates a significant correlation between the knowledge of guidelines and age, professional experience and the level of referral of the hospital. A higher awareness of knowledge and comprehensibility of the protocols was observed in respondents older than 50 years, working longer than 11 years, and in secondary referral departments. The deficit of procedures concerning the prevention, therapy and evaluation of pain in intensive care units has an impact on the quality of analgesic therapy in preterm children and other neonates.

Nowadays, many evidence-based guidelines are available for medical personnel. According to Balice-Bourgois et al. [9], apart from pharmacological and non-pharmacological methods, adequate measures in pain prophylaxis include cooperation with parents [33]. Respondents claim that parents expect pain prevention for their offspring and are aware that pain experienced by the child has an undoubtful impact on the emotional status of the parents. More than 67% of the studied subjects stated that parents should take an active part during invasive procedures. Most subjects participating in the studies carried out by Cong et al. [20] and Mehrnoush et al. [24] claimed that parents expect pain relief for their child and that the pain their child experiences results in stress. Professionals state that parents should take part in actions improving their child’s comfort, especially when painful invasive procedures are necessary. It was proved that the support provided by medical personnel by engaging parents in neonatal pain relief restores and strengthens the ties between them and improves guardians’ trust in the therapeutic methods used in the given department [37,38]. The analysis of the quantitative data shows a significant correlation between family and cultural aspects and education and the level of referral of the center in which the study was carried out. The family and cultural perspective was top-ranked among respondents employed in tertiary referral centers, but was the least important among subjects with secondary education. The introduction of this strategy requires the will to cooperate between healthcare professionals and parents. The results of the studies indicate that most parents want to take part in pain prophylactic actions concerning their child [37,39]. On the other hand, the barriers for implementing such an approach include the unwillingness of the medical personnel to have the parents present during the procedures and a lack of adequate infrastructure for the realization of this strategy [37,39].

The respondents mentioned a lack—or insufficient—amount of training, haste and a lack of staff, unwillingness to change current practices, incorrect habits, incorrect communication within the team, and a lack of belief in the reliability of pain intensity assessment tools as the most important reasons for the introduction of effective pain prophylaxis and therapy in intensive care units.

These results were confirmed by other authors, i.e., Cong et al. [20] and Byrd [40]. The analyses carried out by Cong et al. [20,21,41] seem to confirm the reports presented by other authors and extend the list of difficulties by: unwillingness to introduce changes in everyday practices, knowledge deficit, shortage of time and lack of trust in pain assessment scales/tools.

### Study Limitations

The inclusion of only nurses and midwives in our study is one of its limitations, since the issue of pain applies to every member of the therapeutic team, including physicians. The above-mentioned results refer to the opinions and knowledge about pain and the scales used to assess it presented by the medical personnel. The study represents a quantitative piece of research with no control group, which makes it impossible for us to compare the obtained results.

Another limitation is the number of facilities participating in the project. The invitation to participate in the study was sent to 100 facilities, whereas 52 gave their consent and only 43 took an active part in the study. Participation was voluntary and as our experience shows, the receptiveness and the management’s consent was not associated with the complete engagement of the medical personnel, who often refused to take part in the study.

Despite multiple problems, the project was completed and more than 700 questionnaires were returned, of which more than 500 were subjected to further statistical analysis. Therefore, we may conclude that the obtained results are real and reflect the standpoint of medical personnel concerning the prophylaxis, therapy and monitoring of neonatal pain in Polish intensive care units.

## 5. Conclusions

Our analyses indicate that nurses and midwives obtained knowledge regarding neonatal pain. According to the study subjects, pharmacological and non-pharmacological interventions undertaken in neonatal patients are effective in pain relief. Our study demonstrated that despite the availability of pain assessment tools in neonatal patients, only a few centers use standardized tools. The introduction of strategies to promote and extend the awareness among personnel of the neonatal pain monitoring scales is necessary.

### Implications for Practice

Further studies concerning the control of neonatal pain evaluation will allow the development of adequate training programs in this field. Education on the available scales and the unification of tools for pain assessment in preterm children and neonates born at term would support nurses and midwives in using pain assessment tools in routine clinical practice. It will also enable the exchange of experience and practices between facilities, which would result in creating long-term strategies for pain prophylaxis and therapy in neonatal patients.

## Figures and Tables

**Table 1 healthcare-09-00048-t001:** Socio-demographic characteristics.

Characteristics of the Study Group	*n*	*%*
Gender		
Female	552	98.9
Male	6	1.1
Age		
Less than 30 years	100	17.9
31–50 years	244	43.7
More than 50 years	214	38.4
Place of residence		
Up to 50 thousand inhabitants	225	40.3
50–100 thousand inhabitants	122	21.9
Over 100 thousand inhabitants	211	37.8
Education level		
MSc in nursing/midwifery, specialist in neonatal nursing	178	31.9
BSc in nursing/midwifery	171	30.6
Secondary (medical school of nursing or medical college)	209	37.5
Total professional experience		
Up to 5 years	83	14.9
6–10 years	54	9.7
More than 10 years	421	75.4
Professional experience in department of neonatology		
Less than 1 year	42	7.5
1–10 years	154	27.6
More than 10 years	362	64.9
Referral level of the hospital		
Secondary	340	60.9
Tertiary	218	39.1

**Table 2 healthcare-09-00048-t002:** Nurses’ knowledge and perceptions of neonatal pain.

Questions		M *	SD
Knowledge and Beliefs about Neonatal Pain	Neonates are capable of experiencing pain.	4.66	0.61
	Pain in the neonatal period has long-term adverse effects.	4.04	0.90
	Neonates, especially preterm infants are less sensitive to pain than older children and adults because of the immature immune system.	3.9	1.14
	Neonates will remember painful experiences.	3.26	1.02
	Minor procedures, such as gavage tube placement and oral suctioning can cause pain.	4.01	0.81
	Preterm neonates are at a greater risk of neuro-developmental impairment due to repeated painful procedures in their NICU stay.	3.71	1.03
Nurses’ Perceptions of Pain Assessment Tool in the Neonatal Intensive Care Unit	I received adequate training regarding neonatal pain recognition/assessment when I was oriented to my unit.	2.24	2.49
	My unit provides continuing education regarding neonatal pain management.	2.91	1.14
	My unit uses a neonatal pain assessment tool regularly.	2.67	1.06
	I feel confident with my skills in recognizing and assessing the physiologic/behavioral indicators of neonatal pain.	2.86	1.22
	I feel confident in my use of the neonatal pain assessment tool in my unit.	3.49	0.93
	The pain assessment tool in my unit is an accurate measure of neonatal pain.	2.76	1.03
	I am confident in my ability to interpret scores obtained from pain assessment tools.	2.89	1.01
	I have attended an education conference/lecture in the past 5 years that has included information on neonatal pain.	2.69	1.23
Nurses’ Perceptions of Pain Interventions in the Neonatal Intensive Care Unit	I feel that neonatal pain in my unit is well managed.	2.82	1.09
	It is the responsibility of nurses to advocate in pain management in neonates.	4.03	0.73
	The doctors in my unit prescribe adequate PRN (pro re nata) medicines for pain management.	3.58	0.95
	I feel apprehensive when giving neonates opiates due to the risk of addiction.	2.69	0.93
	I am aware of the pharmacological treatments available for neonatal pain.	3.83	0.69
	Non-pharmacological pain management is effective to manage neonatal pain.	3.60	0.87
	Pharmacological/non­pharmacological interventions are not necessary because many invasive procedures can be done quickly.	2.14	0.89
	I feel confident with my skills in pain management using non-pharmacological interventions in neonates.	3.57	0.84
	I feel confident with my skills in pain management using pharmacological interventions in neonates.	3.55	0.78
Nurses’ Perceptions of Guidelines/Protocols in Pain Management in the NICU	I am aware of the neonatal pain management guidelines/protocols.	3.17	0.96
	The pain management guidelines/protocols are clear, comprehensive, and based on current research on the unit in which I work.	3.04	0.99
	Physicians are willing to utilize new evidence-based practices of pain management in my unit.	2.99	1.02
	I feel confident in my ability to initiate change in relation to pain management in my unit.	2.99	0.95
	I feel that I am comfortable approaching a physician to discuss necessary pain management for my neonatal patient.	3.10	1.05
	I feel comfortable implementing non-pharmacological nursing interventions when I feel a neonate’s pain is not adequately managed.	3.65	0.86
Family/Culture Perspectives in Pain Management	The prevention of pain in neonates is an expectation of parents.	4.08	0.72
	Parents should be involved in the care and comfort of their infant during painful procedures.	3.63	1.02
	Parents are emotionally affected by the pain their infant may be experiencing.	4.36	0.57
	Kangaroo care is an effective non-pharmacological approach to neonatal pain management.	4.14	0.69
	Oral sucrose or a similar alternative is an effective non-pharmacological approach to neonatal pain management.	3.99	0.71

Legend: M *—mean value, the score ranges from 1 to 5. SD—standard deviation. NICU: neonatal intensive care units.

**Table 3 healthcare-09-00048-t003:** A detailed correlation analysis for variables: knowledge and beliefs about neonatal pain, nurses’ perceptions of pain assessment tool in the neonatal intensive care unit, nurses’ perceptions of pain interventions in the neonatal intensive, nurses’ perceptions of guidelines/protocols in pain management in the NICU, family/culture perspectives in pain management.

**Knowledge of neonatal pain vs. age of the medical staff**	***n***	**M**	**SD**	**H**	**df**	***p***
Less than 30 years	100	3.67	0.45	1.40	2	0.496
31–50 years	244	3.63	0.48			
More than 50 years	214	3.62	0.51			
**Nurses’ Perceptions of Pain Assessment Tool in the Neonatal Intensive Care Unit vs. age of the medical staff**	***n***	**M**	**SD**	**H**	**df**	***p***
Less than 30 years	100	2.80	0.89	15.79	2	**0.001**
31–50 years	244	2.69	0.90			
More than 50 years	214	**3.02**	0.88			
**Nurses’ Perceptions of Pain Interventions in the Neonatal Intensive vs. age of the medical staff**	***n***	**M**	**SD**	**H**	**df**	***p***
Less than 30 years	100	3.28	0.45	30.71	2	**0.001**
31–50 years	244	3.23	0.39			
More than 50 years	214	3.42	0.37			
**Nurses’ Perceptions of Guidelines/Protocols in Pain Management in the NICU vs. age of the medical staff**	***n***	**M**	**SD**	**H**	**df**	***p***
Less than 30 years	100	3.03	0.68	29.69	2	**0.001**
31–50 years	244	3.05	0.67			
More than 50 years	214	3.34	0.71			
**Family/Culture Perspectives in Pain Management vs. age of the medical staff**	***n***	**M**	**SD**	**H**	**df**	***p***
Less than 30 years	100	3.90	0.38	1.14	2	0.563
31–50 years	244	3.84	0.48			
More than 50 years	214	3.88	0.47			
**Knowledge of neonatal pain vs. education level of medical staff**	***n***	**M**	**SD**	**F**	**df**	***p***
MSc in nursery/midwifery, specialist in neonatological nursing	178	3.74	0.45	10.45	2.555	**0.001**
BSc in nursery/midwifery	171	3.76	0.46			
Secondary (medical school of nursing or medical college)	209	3.52	0.51			
**Nurses’ Perceptions of Pain Assessment Tool in the Neonatal Intensive Care Unit vs. education level of medical staff**	***n***	**M**	**SD**	**F**	**df**	***p***
MSc in nursery/midwifery, specialist in neonatological nursing	178	2.76	0.94	1.50	2.555	0.223
BSc in nursery/midwifery	171	2.93	0.96			
Secondary (medical school of nursing or medical college)	209	2.82	0.81			
**Nurses’ Perceptions of Pain Interventions in the Neonatal Intensive education level of medical staff**	***n***	**M**	**SD**	**F**	**df**	***p***
MSc in nursery/midwifery, specialist in neonatological nursing	178	3.28	0.42	0.85	2.555	0.428
BSc in nursery/midwifery	171	3.32	0.45			
Secondary (medical school of nursing or medical college)	209	3.33	0.35			
**Nurses’ Perceptions of Guidelines/Protocols in Pain Management in the NICU vs. education level of medical staff**	***n***	**M**	**SD**	**F**	**df**	***p***
MSc in nursery/midwifery, specialist in neonatological nursing	178	3.08	0.77	1.69	2.555	0.185
BSc in nursery/midwifery	171	3.19	0.67			
Secondary (medical school of nursing or medical college)	209	3.20	0.66			
**Family/Culture Perspectives in Pain Management vs. education level of medical staff**	***n***	**M**	**SD**	**F**	**df**	***p***
MSc in nursery/midwifery, specialist in neonatological nursing	178	3.96	0.47	7.46	2.555	**0.001**
BSc in nursery/midwifery	171	3.86	0.47			
Secondary (medical school of nursing or medical college)	209	3.68	0.44			
**Knowledge of neonatal pain vs. professional experience in neonatological department**	***n***	**M**	**SD**	**H**	**df**	***p***
Less than 1 year	42	3.63	0.56	1.36	2	0.507
1–10 years	154	3.59	0.47			
More than 10 years	362	3.65	0.49			
**Nurses’ Perceptions of Pain Assessment Tool in the Neonatal Intensive Care Unit vs. professional experience in neonatological department**	***n***	**M**	**SD**	**H**	**df**	***p***
Less than 1 year	42	2.97	1.07	3.28	2	0.193
1–10 years	154	2.72	0.77			
More than 10 years	362	2.87	0.93			
**Nurses’ Perceptions of Pain Interventions in the Neonatal Intensive vs. professional experience in neonatological department**	***n***	**M**	**SD**	**H**	**df**	***p***
Less than 1 year	42	3.35	0.53	5.00	2	0.082
1–10 years	154	3.25	0.34			
More than 10 years	362	3.33	0.41			
**Nurses’ Perceptions of Guidelines/Protocols in Pain Management in the NICU vs. professional experience in neonatological department**	***n***	**M**	**SD**	**H**	**df**	***p***
Less than 1 year	42	3.13	0.71	10.52	2	**0.005**
1–10 years	154	3.03	0.57			
More than 10 years	362	3.22	0.74			
**Family/Culture Perspectives in Pain Management vs. professional experience in neonatological department**	***n***	**M**	**SD**	**H**	**df**	***p***
Less than 1 year	42	3.94	0.40	2.55	2	0.279
1–10 years	154	3.84	0.43			
More than 10 years	362	3.87	0.48			
**Knowledge of neonatal pain vs. referral level of the hospital**	***n***	**M**	**SD**	t	**df**	***p***
Level II NICU	340	3.56	0.51	3.96	555	**0.001**
Level III NICU	218	3.72	0.43			
**Nurses’ Perceptions of Pain Assessment Tool in the Neonatal Intensive Care Unit vs. referral level of the hospital**	***n***	**M**	**SD**	t	**df**	***p***
Level II NICU	340	2.88	0.89	1.53	555	0.126
Level III NICU	218	2.76	0.90			
**Nurses’ Perceptions of Pain Interventions in the Neonatal Intensive vs. referral level of the hospital**	***n***	**M**	**SD**	t	**df**	***p***
Level II NICU	340	3.43	0.38	1.96	555	**0.050**
Level III NICU	218	3.27	0.42			
**Nurses’ Perceptions of Guidelines/Protocols in Pain Management in the NICU vs. referral level of the hospital**	***n***	**M**	**SD**	t	**df**	***p***
Level II NICU	340	3.22	0.67	2.70	555	**0.007**
Level III NICU	218	3.05	0.72			
**Family/Culture Perspectives in Pain Management vs. referral level of the hospital**	***n***	**M**	**SD**	t	**df**	***p***
Level II NICU	340	3.83	0.45	2.08	555	**0.037**
Level III NICU	218	3.91	0.46			

Legend: *p* value < 0.05 (bold); M—mean; SD—standard deviation; df—distribution free; H—statistic; t—Student’s *t*-test.

## Data Availability

The data that support the findings of this study are available from the corresponding author, (H.P), upon reasonable request.

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
