# Peer review of "Knowledge and Practices in Neonatal Pain Management of Nurses Employed in Hospitals with Different Levels of Referral—Multicenter Study"

_healthcare, 2021, doi:10.3390/healthcare9010048_

Round 1
Reviewer 1 Report
Dear authors,
Thank you very much for your paper that highlights a very common problem in neonatal pain assessment, prevention and treatment. There is an overall agreement that many issues in this regard have to be improved.
I have however some major and minor comments to improve the quality and credibility of your work.
Looking forward receiving the revised version
Sincerely yours
Reviewer
Major comments:
Did the study protocol require to be approved by IRB ? Please provide this information. Was this study submitted for clinical trial registration?
Minor comments:
Line 49. Which word stands for APP? Please provide the complete word at the first time that an abbreviation is used.
Please provide an abbreviation list.
Line 62. I think in this line the authors meant respiratory tract suctioning. Please replace upper respiratory tract suction by respiratory tract suctioning.
Line 81. This sentence (The aim of this study was to assess the of knowledge and the perception of the issue of pain in ...) is a difficult to understand. Please modify and simplify.
Line 85. This sentence (It was a quantitative descriptive study, which was carried out in Polish neonatal intensive care units) is difficult to understand, please modify.
Table 2. Something is written at the left side of table but it is partially hidden behind the table and is not readable.
Table 2. In order to understand better the mean and SD at the table, perhaps it would be sensible to add a very small text in the column of M that the scores are out of 5.
I think it is difficult to conclude from means of a score that the majority or minority of the respondents could be agree or disagree with a certain point of view, unless this information is obviously associated with the numbers of respondents. For example, in a score out of 20, in the below mentioned scores, the means of both scores are 10. However, the majority in first population scored below 10 whereas the majority in second population seems to be acceptable. In this case first population deserves more attention.
6, 6, 8, 20; mean = 10
4, 10, 10, 16; mean = 10
Line 157. I’m not sure from where the number of 2.1 comes? At least if I am correct, perhaps it would be more sensible to state that “Although a significant number of the study group ?? (if this is true) realizes that neonates will remember their painful experience, the majority of those ?? (if this is true) also believe that neonates, and especially preterm infants are less sensitive to pain than older children and adults because of the immature immune system”.
Results section
This section may be shortened in max 3 pages including the tables.
Some parts of less importance should be explained briefly.
Line 244. This sentence is difficult to understand “The results clearly indicate that the although the awareness of most studied respondents is good, there are neither adequate system solutions nor unified assessment tools in healthcare facilities available, which requires improvement”. Please modify.
Discussion section
Some sentences are difficult to understand and this is partially related to wordings in English language, I think.
The most conclusions obtained in this study were already demonstrated by other authors. Can authors of this study identify the new and unique issues related to this study?
Author Response
|
Did the study protocol require to be approved by IRB ? Please provide this information. Was this study submitted for clinical trial registration? Dear Review, thank you for your suggestion. The research was conducted with the use of a descriptive method. The project was approved by the Independent Bioethical Committee for Scientific Research of the Medical University of Gdansk (approval number: NKBBN/116/2019). The nature of the project did not meet the criteria of Clinical Trail registration. This was the main reason why the study was not submitted to that path. Which word stands for APP? Please provide the complete word at the first time that an abbreviation is used. Line 49- The abbreviation was added to the text and at the end of the manuscript. “American Academy of Pediatrics (APP) & Canadian Pediatric Society (CPS) , and Polish Neonatal Society”. The abbreviation was added in line 349. I think in this line the authors meant respiratory tract suctioning. Please replace upper respiratory tract suction by respiratory tract suctioning. Line 62 - The phrase was changed into the respiratory tract suctioning. This sentence (The aim of this study was to assess the of knowledge and the perception of the issue of pain in ...) is a difficult to understand. Please modify and simplify. Line 81- It is not possible to reduce or change the sentence. The main aim of the study was to evaluate the knowledge and the attitude of the medical staff in the field of pain management and prophylaxis. The original version of the questionnaire was constructed to focus on 5 aspects related to the perception of the neonatal pain by nurses: (1) knowledge and beliefs; (2) the usage of assessment tools; (3) the usage of pharmacologic and nonpharmacologic interventions; (4) guidelines/protocols and family involvement; and (5) barriers and strategies. This sentence (It was a quantitative descriptive study, which was carried out in Polish neonatal intensive care units) is difficult to understand, please modify. Line 85- The sentence was changed into: “The research was conducted with the usage of a descriptive method. The project was carried out in the Polish neonatal intensive care units.” Something is written at the left side of table but it is partially hidden behind the table and is not readable. Table 2.- The table was moved and the information is visible now. In order to understand better the mean and SD at the table, perhaps it would be sensible to add a very small text in the column of M that the scores are out of 5. Table 2.The information was added into the text. The score ranges from 1 to 5. I think it is difficult to conclude from means of a score that the majority or minority of the respondents could be agree or disagree with a certain point of view, unless this information is obviously associated with the numbers of respondents. For example, in a score out of 20, in the below mentioned scores, the means of both scores are 10. However, the majority in first population scored below 10 whereas the majority in second population seems to be acceptable. In this case first population deserves more attention. 6, 6, 8, 20; mean = 10 4, 10, 10, 16; mean = 10 Thank you for the important suggestion. It was a statistical misunderstanding. The data was corrected. I’m not sure from where the number of 2.1 comes? At least if I am correct, perhaps it would be more sensible to state that “Although a significant number of the study group ?? (if this is true) realizes that neonates will remember their painful experience, the majority of those ?? (if this is true) also believe that neonates, and especially preterm infants are less sensitive to pain than older children and adults because of the immature immune system”. -Line 157 -Thank you for the important suggestion. It was a statistical misunderstanding. The proper result should be 3.9. This result is presented in the table no 2. Although a significant number of the study group (M 4.66; SD± 0.61) realizes that neonates will remember their painful experience, the majority of those (M 3.71; SD ±1.03) also believe that neonates, and especially preterm infants are less sensitive to pain than older children and adults because of the immature immune system This sentence is difficult to understand “The results clearly indicate that the although the awareness of most studied respondents is good, there are neither adequate system solutions nor unified assessment tools in healthcare facilities available, which requires improvement”. Please modify. Line 244 - The sentence was changed to: “The results clearly show that the majority of the respondents have a sufficient level of the knowledge of pain. However, in most of the hospitals there are no proper solutions related to the pain assessment tools. This is definitely an area for future improvement”. Some sentences are difficult to understand and this is partially related to wordings in English language, I think. The most conclusions obtained in this study were already demonstrated by other authors. Can authors of this study identify the new and unique issues related to this study? The sentence was changed into: Our results were confirmed by another European researcher who used the same original version of the questionnaire. In Poland the project has been the first one of this kind not only because it is focused on the problem of the pain in the neonatal unit, but also because the research was dedicated to the medical staff who is involved in the pain management. |
Reviewer 2 Report
The authors are trying to assess the medical personnel's knowledge and pain perception in neonatal patients. Authors have used a modified Nurses' Perceptions of Neonatal Pain questionnaire.
Overall the manuscript is written reasonably well, but I have a few concerns/suggestions.
In the abstract: the results section was incomplete.
In line 115,116, it is better to include why the other facilities did not participate in the study.
In line 120, consider rephrasing the inclusion criteria. The authors have mentioned no exclusion criteria.
From line 173 " Unfortunately, more than 55.2% of the studied subjects (n=308) does not use any of the recommended tools for pain monitoring in this group of patients,"
I would recommend doing a multivariate logistic regression with respondents' characteristics to see which variable would be at risk for not using the tool.
I would recommend the same for " guidelines/protocols concerning pain management" as at least 60% of them did not know about the guidelines.
From line 207: what do they mean by bad habits? (caution in choosing the term, as this may not be a politically correct word)
From line 216, authors can present variables associated with knowledge and opinion about pain relief and monitoring by nurses/midwives in a table.
In line 256, did the authors mean secondary referral centers rather than tertiary as their results contradict the above paragraph?
There is no mention of any strengths or limitations of the study.
Line 362-372 can be better represented in the table as a barrier to medical personnel's knowledge and pain perception in neonatal patients.
Author Response
It is better to include why the other facilities did not participate in the study.
Line 115-116- The line was extended with the information
„ Out of 52 hospitals only 43 were included in the study procedure although all of the medical centers agreed to participate in the project. The questionnaire’s package was delivered to all the medical facilities which gave the official consent. In some of the facilities medical staff refused to participate in the researcher. That is why those hospitals were not part of the further analysis.”
Consider rephrasing the inclusion criteria. The authors have mentioned no exclusion criteria.
Line 120-The inclusion and exclusion criteria were modified and added to the text.
" Unfortunately, more than 55.2% of the studied subjects (n=308) does not use any of the recommended tools for pain monitoring in this group of patients,"
I would recommend doing a multivariate logistic regression with respondents' characteristics to see which variable would be at risk for not using the tool.
I would recommend the same for " guidelines/protocols concerning pain management" as at least 60% of them did not know about the guidelines.
Line – 173 Dear Reviewer, thank you for your comment. We think that this issues is very important. However, due to the fact that it is so significant it deserves to be explained in the next manuscript. The aim of our study was mainly to assess the knowledge and attitude of the medical staff in the field of pain. We are aware of the fact that the article will not cover all issues of pain in every detail and that is why we would continue our work and research in that field.
what do they mean by bad habits? (caution in choosing the term, as this may not be a politically correct word)
Bad habits was changed into the bad practices
authors can present variables associated with knowledge and opinion about pain relief and monitoring by nurses/midwives in a table.
From the line 216 - The table 3 was added into the text
did the authors mean secondary referral centers rather than tertiary as their results contradict the above paragraph?
From line 256- The sentence in the line was changed to
“Our results demonstrated that the respondents older than 50 and those employed in the tertiary referral centers presented a good level of neonatal pain knowledge. However, the medical staff of a secondary referral care have a higher level of knowledge in the area of guidelines and protocol when compared with the staff from tertiary hospitals.
There is no mention of any strengths or limitations of the study.
The section of the study limitation was added:
“The inclusion of only nurses and midwives in our study is one of its limitations, since the issue of pain applies to every member of the therapeutic team, including physicians. The above-mentioned results refer to the opinions and knowledge about pain and the scales used to assess it presented by the medical personnel. It was a quantitative and research study with no control group, which makes it impossible for us to compare the obtained results.
Another limitation is the number of facilities participating in the project. The invitation to participate in the study was sent to 100 facilities, whereas 52 gave their consent and only 43 took an active part in the study. Participation was voluntary and as our experience shows, the receptiveness and the management’s consent was not associated with the complete engagement of the medical personnel, who often refused to take part in the study.
Despite multiple problems, the project was completed and more than 700 questionnaires were returned of which more than 500 were subjected to further statistical analysis. Therefore, we may conclude that the obtained results are real and reflect the standpoint of medical personnel concerning the prophylaxis, therapy and monitoring of neonatal pain in Polish intensive care units.”
Can be better represented in the table as a barrier to medical personnel's knowledge and pain perception in neonatal patients.
Line 362-372 - Dear Reviewer, thank you for your valuable comment. Our research is not focused only on the problem of the barriers but mainly on knowledge and the attitude. We are working now on the article which would explain the problem of the barriers among the medical staff in the pain management. Our research showed that the main barrier in the field of pain is lack of educational programs for employees and lack of support from the medical centers.
Reviewer 3 Report
Popowicz et al. describe results of a multiinstitutional national survey of neonatal nurses on assessment and management of pain in neonatal intensive care units. The topic is germane in the management of critically ill infants who cannot verbalize discomfort as the authors mention. Research ethical review was performed by an institutional review board, but it is unclear whether participants consented to take part in the study. The manuscript has some grammatical issues and would benefit from additional proofreading to assist fluidity and readability.
The overall text is verbose and can be condensed considerably to focus on the primary findings.
The introduction can be shortened and needs to focus on the assessment and management of pain. The text from L33-47 can be removed and the paper should start with the issue of pain in the NICU. Multiple statements need supporting citations including what statements have been made by the medical societies mentioned on L50-1, as well as relevant studies to support statements on L52, 57, 71, 76. For the inadequate implementations of pain intervention mentioned on L79, can the authors discuss relevant outcomes related to these issues?
Painful procedures on L61-2 should add “in addition to others” so as to include unspecified painful procedures such as ophthalmological exams, arterial puncture, etc.
On L102 can the authors expand on why race and ethnicity were removed from the questionnaire? Can they describe the prevalence of relevant ethnicities in Poland as the reason for excluding these questions is not clear.
In the statistical analysis Bonferroni adjustment is described but a P≤0.05 was stated as significant, can the authors clarify.
Overall reporting of the results is too verbose and interpretation can be reserved for the discussion.
On L173 please delete “unfortunately,” and on L174 “any of the”, and clarify the source of the recommendation.
On 177 respondents averaged high agreement with the opinion that they should promote pain issues, not that a great majority held the opinion.
The discussion is simply too long and traverses a wide variety of topics. I recommend that the authors reduce the text by a third to a half and focus on action items including increasing education and use of pain assessment tools as well as other conclusions. Opinions and judgements should also be reduced or qualified with accompanying data or citations. Use of the word “unsatisfactory” should be reduced, as the question arises as to whom is the issue unsatisfactory? “Contradictory” on L275 and L334 don’t seem to be a good choice of words, the percentage is merely different.
The sentence on L362-5 is hard to understand. Were these reasons for ineffectiveness?
In summary there are important conclusions to make from this data in terms of pain assessment and the need for more formalization of its methods. The scope of this survey is important and its results are valuable in that regard. It would help to reduce the amount of text and focus on significant findings in the study as well as reasons for them. Consent from the respondents or an IRB waiver of consent should be described.
Author Response
The introduction can be shortened and needs to focus on the assessment and management of pain. The text from L33-47 can be removed and the paper should start with the issue of pain in the NICU.
from line 33-47 Thank you for the opinion. However, it is difficult to relate to it, because the opinions of other reviewers are quite diversified.
Multiple statements need supporting citations including what statements have been made by the medical societies mentioned on L50-1, as well as relevant studies to support statements on L52, 57, 71, 76.
The reference were added in the text
For the inadequate implementations of pain intervention mentioned on L79, can the authors discuss relevant outcomes related to these issues?
L79The text was extended
„Other researchers demonstrate multiple inadequate implementations of the guidelines in terms of pain therapy. These include lack of continuous and regular pain intensity assessment, incomplete introduction of non-pharmacological methods of pain relief and low efficiency of pain therapyhis may cause psychological, behavioral, metabolic, mental and hormonal reactions.
Painful procedures on L61-2 should add “in addition to others” so as to include unspecified painful procedures such as ophthalmological exams, arterial puncture, etc.
Line 61-62The sentence was changed into: Neonates experience average of 12-16 medical procedures per day, most cause mild or moderate pain. Apart from them it is worth to mention they include heel prick, venous catheter insertion, venous blood collection, respiratory tract suctioning or intubation.
can the authors expand on why race and ethnicity were removed from the questionnaire? Can they describe the prevalence of relevant ethnicities in Poland as the reason for excluding these questions is not clear.
Line 102Thank you for this review. In Poland we have to exclude the question about ethnicity. National Statistic Office in Poland doesn’t have such a diversity when it comes to nationality when compared to other European countries. Most of the population is Polish. Reference: Statistics Poland. Address:https://stat.gov.pl/spisy-powszechne/nsp-2011/nsp-2011-wyniki/ludnosc-stan-i-struktura-demograficzno-spoleczna-nsp-2011,16,1.html
In the statistical analysis Bonferroni adjustment is described but a P≤0.05 was stated as significant, can the authors clarify.
The Kurskal-Wallis test showed the significant difference between studied groups. the analysis with the usage of Bonferroni test showed the difference between groups being compared. To show the difference p<0.05 was used.
Overall reporting of the results is too verbose and interpretation can be reserved for the discussion.
Thank you for the opinion. However, it is difficult to relate to it, because the opinions of other reviewers are quite diverse.
On L173 please delete “unfortunately,” and on L174 “any of the”, and clarify the source of the recommendation
Line 173The changes were There is the information about the sources of recommendations in the Introduction section L71-73.
On 177 respondents averaged high agreement with the opinion that they should promote pain issues, not that a great majority held the opinion.
Line 177The sentence was revised and changed to:
The significant number of the researched group (M 4.03; SD± 0.73) held the opinion that it is nurses/midwives’ duty to promote issues associated with pain prophylaxis and management in their department. Some respondents (M 2.82; SD± 1.09) did not feel adequately trained by their employer in both pain prophylaxis and forms of its management in paediatric patients.
The discussion is simply too long and traverses a wide variety of topics. I recommend that the authors reduce the text by a third to a half and focus on action items including increasing education and use of pain assessment tools as well as other conclusions. Opinions and judgements should also be reduced or qualified with accompanying data or citations. Use of the word “unsatisfactory” should be reduced, as the question arises as to whom is the issue unsatisfactory? “Contradictory” on L275 and L334 don’t seem to be a good choice of words, the percentage is merely different.
Thank you for the opinion. However, it is difficult to refer to it, because of the first reviewer recommended to expand the discussion section . We have revised it once again and reduced words/phrases errors The Line L.275-L334 was changed to
Cong et al. [16], presented completely different results. The author with his research team stated that more than 80% of nurses used pain assessment tools. All respondents who used pain assessment scales in their routine practice were confident in their use and interpretation in neonatological patients [16].
In summary there are important conclusions to make from this data in terms of pain assessment and the need for more formalization of its methods. The scope of this survey is important and its results are valuable in that regard. It would help to reduce the amount of text and focus on significant findings in the study as well as reasons for them. Consent from the respondents or an IRB waiver of consent should be described.
Thank you for the opinion. However, it is difficult to refer to it, because the opinions of other reviewers are quite different The text has been revised once again, we tried to take all the suggestions into consideration. It is very difficult to reduce the amount of text when some of the reviewers suggest to expand the manuscript in the field of results or discussion.
The sentence on L362-5 is hard to understand. Were these reasons for ineffectiveness?
This was changed into: On the other hand, the barriers for implementing such approach include unwillingness of the medical personnel to the presence of the parents during the procedures and lack of adequate infrastructure for the this strategy realization.The respondents mentioned lack or insufficient number of trainings, haste and lack of staff, unwillingness to change current practice, incorrect habits, incorrect communication within the team, lack of belief in the reliability of pain intensity assessment tools as the most important reasons for effective introduction of effective pain prophylaxis and therapy in intensive care units.
Round 2
Reviewer 2 Report
Authors can consider poor or inappropriate practice instead of bad practice.
Author Response
Authors can consider poor or inappropriate practice instead of bad practice.
Resonse
It’s correct, line 207
Reviewer 3 Report
I appreciate the authors' responses to my comments. Unfortunately I do not have access to the other reviewers' comments so I do not understand the conflicts with my comments that were not reconciled. I still believe the introduction, results, and discussion could be more focused because at present these sections appear too dense. I defer to the editors, since I cannot see the other reviewers' comments.
My question about ethnicity could be clarified in the text. The the text on line 173 should include references to what the recommended tools are, it is not clear that the authors mean to refer to lines 71-73.
The statement of Cong et al now on Line 349 still uses the word contradictory which makes it unclear whether the paper contradicts this one or appears contradictory within the paper. Please change "Completely contradictory" to "Different".
Author Response
I appreciate the authors' responses to my comments. Unfortunately I do not have access to the other reviewers' comments so I do not understand the conflicts with my comments that were not reconciled. I still believe the introduction, results, and discussion could be more focused because at present these sections appear too dense. I defer to the editors, since I cannot see the other reviewers' comments.
Resonse
deleted text that was repeated so as not to change the context of the work
My question about ethnicity could be clarified in the text. The the text on line 173 should include references to what the recommended tools are, it is not clear that the authors mean to refer to lines 71-73.
Resonse
It’correct, line 97-99, reference no. 21 has been added (line 466-469)
It’correct, line 170 - references added [15,16]
The statement of Cong et al now on Line 349 still uses the word contradictory which makes it unclear whether the paper contradicts this one or appears contradictory within the paper. Please change "Completely contradictory" to "Different".
Resonse
It’s correct, line 335